# Ternary Mixed Micelle Hexadecyltrimethylammonium Bromide—Dodecyltrimethylammonium Bromide—Sodium Deoxycholate: Gibbs Free Energy of Mixing and Excess Gibbs Energy of Mixing

**DOI:** 10.3390/molecules28186722

**Published:** 2023-09-20

**Authors:** Ana Pilipović, Ivana Vapa, Vesna Tepavčević, Gorana Puača, Mihalj Poša

**Affiliations:** Department of Pharmacy, Faculty of Medicine, University of Novi Sad, Hajduk Veljka 3, 21000 Novi Sad, Serbia; ivana.vapa@gmail.com (I.V.); vesna.tepavcevic@mf.uns.ac.rs (V.T.); gorana.puaca@mf.uns.ac.rs (G.P.)

**Keywords:** critical micelle concentration, surfactants, thermodynamic stabilization, conformations, mixed micelles, regular solution theory, bile salts

## Abstract

Pharmaceutical, food, and cosmetic formulations often contain binary or ternary surfactant mixtures with synergistic interactions amongst micellar building blocks. Here, a ternary mixture of the surfactants hexadecyltrimethylammonium bromide, dodecyltrimethylammonium bromide, and sodium deoxycholate is examined to see if the molar fractions of the surfactants in the ternary mixed micellar pseudophase are determined by the interaction coefficients between various pairs of the surfactants or by their propensity to self-associate. Critical micelle concentrations (CMC) of the analyzed ternary mixtures are determined experimentally (spectrofluorimetrically using pyrene as the probe molecule). Thermodynamic parameters of ternary mixtures are calculated from CMC values using the Regular Solution protocol. The tendency for monocomponent surfactants to self-associate (lower value of CMC) determines the molar fractions of surfactant in the mixed micelle if there is no issue with the packing of the micelle building units of the ternary mixed micelle. If a more hydrophobic surfactant is incorporated into the mixed micelle, the system (an aqueous solution of surfactants) is then the most thermodynamically stabilized.

## 1. Introduction

Surfactants are molecules or ions that contain both hydrophilic and hydrophobic regions in their structure. In an aqueous solution, surfactants are embedded in the aqueous solution–air interface and the hydrophobic molecular segment of the surfactant is oriented toward the air; while the hydrophilic polar head is orientated toward the aqueous phase, the Gibbs free energy of the water–surfactant system decreases. On a particular total surfactant concentration, known as the critical micelle concentration (CMC), the air and water phase’s boundary surfaces are completely saturated with surfactant particles. In order for the Gibbs free energy to further decrease in the water–surfactant system, surfactants form micelles, molecular aggregates in which the hydrophobic part of the surfactant forms the hydrophobic core of the micelle, while the polar groups of the surfactants form the polar outer shell of the micelle [1,2,3,4,5]. It has been demonstrated that below the CMC, surfactants form aggregates of a few (2–4) surfactants called premicellar aggregations, which can exhibit catalytic capabilities in certain chemical processes [6,7,8]. Both monocomponent surfactants and a precisely calculated mixture of surfactants can generate micelles, which are known as mixed micelles. Surfactant mixes that are either binary or ternary are typically utilized, and critical micellar concentrations for these mixtures have also been identified (i.e., determined experimentally) [9,10,11,12,13]. According to the phase separation model (applicable for micelles with large aggregation numbers, i.e., the number of surfactant particles in one micelle), mixed micelles (as micellar pseudophases) are formally created by mixing monocomponent micellar pseudophases [14,15,16,17,18]. The molar Gibbs free energy of the formation of a ternary mixed micellar pseudophase from a ternary mixture of surfactants dissolved in the aqueous phase (solution) is (${∆}_{f}g_{mM}^{0}$):(1)$${{∆}_{f}g}_{mM}^{0}=x_{1}{∆}_{f}g_{M1}^{0}+x_{2}{{∆}_{f}g}_{M2}^{0}+x_{3}{{∆}_{f}g}_{M3}^{0}+{∆g}_{mix}^{id}+g_{mix}^{E}$$

In Equation (1), ${∆}_{f}g_{Mi}^{0}=RT\ln{CMC}_{i}\;(i=1,2,3)$ represents the molar Gibbs free energy of the formation of monocomponent micellar pseudophases from which a ternary mixed micellar pseudophase is formally obtained (1 mol); if the critical micelle concentration of the monocomponent (pure) surfactant is ${CMC}_{i}$, $x_{i}$ (i=1,2,3) represents the molar fractions of surfactants in the ternary mixed micelle, i.e., micellar pseudophase; ${∆g}_{mix}^{id}$ corresponds to the change in molar Gibbs free energy when obtaining a mixed micellar pseudophase as an ideal mixture of monocomponent micellar pseudophases under the conditions of constant pressure and temperature (i.e., ideal Gibbs free energy of mixing) [14,15,16]. The last term in Equation (1) presents the excess molar Gibbs free energy and contains energetic (entalpic) and entropic effects (interactions) not present in monocomponent micellar pseudophases [19]:(2)$${{∆}_{f}g}_{mM}^{0}-x_{1}{∆}_{f}g_{M1}^{0}+x_{2}{{∆}_{f}g}_{M2}^{0}+x_{3}{∆}_{f}g_{M3}^{0}=\left\{\begin{array}{c} {∆g}_{mix}^{id}+g_{mix}^{E}=∆g_{mix}^{rel}\\ {∆g}_{mix}^{id} \end{array}\right.$$

The sum ${∆g}_{mix}^{id}+g_{mix}^{E}$ presents the real molar Gibbs free energy of mixing ($∆g_{mix}^{rel}$). Generally for ideal mixtures, $g_{mix}^{E}=0$. In accordance, $g_{mix}^{E}$ describes the thermodynamic stabilization of real mixed micellar pseudophases, related to the ideal mixed micellar pseudophases [16]. According the to Regular Solution Theory (RST), the excess molar Gibbs free energy originates from energetic interactions between the first neighbors of structurally different surfactants, while surfactant conformations in mixed micellar pseudophases are identical to conformations of the same surfactants in monocomponent pseudophases [20,21]. Different types of particles are randomly distributed across the crystal lattice. This means that there is no excess molar entropy in RST (compared to an ideal mixture) [21,22]. The excess molar Gibbs free energy for the binary micellar pseudophase based on RST is expressed by the following symmetric Margules function of the first order: (3)$$g_{mix}^{E}=h_{mix}^{E}=RT\beta_{12}x_{1}x_{2};\;\left.s_{mix}^{E}=0\right\}RST$$
where *R* is the universe gass constant, *T* presents the thermodynamic temperature of the system, while $\beta_{12}$ is the interaction coefficient (interaction parameter) between surfactants 1 and 2 in the binary mixed micellar pseudophase [14,15,16,20,21,22]. The coefficient of the interaction depends on the geometry of binary mixed micellar pseudophase which is represented as a quasi-crystalline structure whose parameter is the coordination number [20,21,22]. If $\beta_{12}<0,$ then there are synergistic interactions between surfactants 1 and 2 ($∆g_{mix}^{rel}<{∆g}_{mix}^{id}$), i.e., the real binary mixed micellar pseudophase is thermodynamically more stable than the ideal binary mixed micellar pseudophase. On the contrary, if $\beta_{12}>0$, there are antagonistic interactions between surfactants 1 and 2, and real mixed micelle is thermodynamically less stable than the ideal mixed micelle ($∆g_{mix}^{rel}>{∆g}_{mix}^{id}$) [16,23,24,25]. According to Porter-u and van Laar, the Margules function of the first order (**3**) when $s_{mix}^{E}\neq 0$ can also describe the excess Gibbs free energy of mixing [26,27,28]. This is the case if the phenomenon of enthalpy–entropy compensation applies to the thermodynamic process of mixed micelle formation. Then, the excess molar Gibbs free energy is a symmetric function of the mole fraction of surfactants from the binary mixed micellar pseudophase [21,22,29,30,31]. In the case of the ternary micellar pseudophase, the excess molar Gibbs energy of mixing is the function of interaction parameters between structurally different surfactant particles: $g_{mix}^{E}=f\left(\beta_{12},\beta_{13,}\beta_{23}\right)\wedge \beta_{ij}=\beta_{ji}$, Therefore, it is necessary to know the interaction parameters in binary mixed micellar pseudophases [11,32,33]. 

Monocomponent, binary, and ternary mixtures of surfactants are usually applied in pharmaceutical, food, and cosmetic formulations; in petro chemistry in the micellar catalysis of chemical reactions/synergistic interactions between surfactants, in addition to the thermodynamic parameters of micellization, they can also occur in different properties of surfactants, such as the solubilization capacity of hydrophobic molecules. If there are synergistic interactions between surfactants, and if $\beta_{ij}$ is negative enough, then the critical micelle concentration of the binary mixture of surfactants can have a lower value than the critical micelle concentration of the more hydrophobic surfactant in the mixture.

This means that to achieve the same (surface) effect from a binary mixture of surfactants, a smaller amount of surfactants is used than from a more hydrophobic surfactant—the ecological footprint of the binary mixture is reduced compared to monocomponent surfactants [4,5,6,7,8,9,10,11,12,16,34,35,36,37].

The aim is to determine whether the interaction coefficients between binary pairs of surfactants ($\beta_{ij}$) or the tendency towards the self-association of monomeric surfactants determines the molar fractions of surfactants in the ternary mixed micellar pseudophase of hexadecyltrimethylammonium bromide (**1**)—dodecyltrimethylammonium bromide (**2**)—sodium deoxycholate (**3**) (Figure 1, Appendix A). Building units of the examined ternary micelle are cationic and anionic surfactants, so it is expected that there are synergistic interactions between cationic and anionic surfactants due to Columb’s electrostatic attractive interactions. The different geometry of the hydrophobic segment of cationic (**1**) (**2**) on one side and anionic (**3**) surfactants on the other side results in the excess of molar (conformational) entropy [38]. The different lengths of the hydrocarbon chains between the examined cationic surfactants (**1**) and (**2**) means that surfactants (**1**) and (**2**) have different tendencies towards self-association, i.e., the cationic surfactant with the longer hydrocarbon chain has the lower value of critical micelle concentration (i.e., greater tendency towards self-association). In the ternary mixture of surfactants, the molar fraction of sodium-deoxycholate is constant ($\alpha_{3}=0.6$), whereas the cationic surfactant ratio changes: $\alpha_{1}+\alpha_{2}+0.6=1$ (the molar fraction of surfactants in the starting ternary mixture of surfactants—which dissolves in the aqueous phase—differs from the molar fraction of the same surfactants in the formed ternary mixed micelles $x_{i}\neq \alpha_{i}$). The mole fraction of sodium deoxycholate (**3**) was chosen based on our earlier investigations into the solubilization capacity of binary mixtures, i.e., for $\alpha_{3}=0.6$ in a binary mixed micelle, it approximately corresponds to $x_{3}\approx 0.5$, which is desirable for the solubilization of flavonoids, isoflavonoids, and polyphenols (bile acid anions form hydrogen bonds with flavonoids and, thus, are incorporated into the hydrophobic core of the mixed micelle) [39].

## 2. Results and Discussion

At some values of $\alpha_{1}+\alpha_{2}+0.6=1$, a reaction of ternary mixed micelle formation is observed (as a summary process of successive and parallel association processes [40], since the tested micelle does not bind counterions, the charge of the micelle was not taken into consideration):(4)$$n_{i}1_{\left(aq\right)}+n_{2}2_{\left(aq\right)}+n_{3}3_{\left(aq\right)}⇄mM{\left\{x_{1}=\frac{n_{1}}{n_{1}+n_{2}+n_{3}},x_{2}=\frac{n_{2}}{n_{1}+n_{2}+n_{3}},\;x_{1}+x_{2}+x_{3}=1\;\right\}}_{\left(aq\right)}$$
where the micelle’s composition and the micelle’s aggregation number correspond to the mean values over the reaction system. The chemical potentials of surfactants, i.e., ternary mixed micelles as separate particles, can be expressed by Henry’s law, where the standard state refers to an infinitely (ideally) diluted solution. Thus, in the equilibrium state for the process (4), the following holds:(5)$${\left(\frac{\partial G}{\partial \xi }\right)}_{p,T}=0=\left(\mu_{mM}^{0}-n_{1}\mu_{1}^{0}-n_{2}\mu_{2}^{0}-n_{3}\mu_{3}^{0}\right)+RT\left(\ln x_{mM}^{b}-n_{1}\ln x_{1}^{b}-n_{2}\ln x_{2}^{b}-n_{3}\ln x_{3}^{b}\right)$$

In the above equation, $\mu_{mM}^{0}$ is the standard chemical potential of the ternary mixed micelles in an aqueous solution. At the same equation, $\mu_{i}^{0}$ (i=1,2,3) represents the standard chemical potential of a surfactant (monomer) in an aqueous solution; $x_{mM}^{b}$ and $x_{i}^{b}$ (i=1,2,3) correspond to the molar fractions of the mixed micelles and surfactants in an aqueous solution. Multiplying Equation (5) by the reciprocal of the aggregation number of the ternary mixed micelle ($1/\left(n_{1}+n_{2}+n_{3}\right)$ and $n=n_{1}+n_{2}+n_{3}$), we obtain the equation:(6)$$\frac{-\left(\mu_{mM}^{0}-n_{1}\mu_{1}^{0}-n_{2}\mu_{2}^{0}-n_{3}\mu_{3}^{0}\right)}{n}=+RT\left(\ln{\left(x_{mM}^{b}\right)}^{\frac{1}{n}}-\frac{n_{1}}{n}\ln x_{1}^{b}-\frac{n_{2}}{n}\ln x_{2}^{b}-\frac{n_{3}}{n}\ln x_{3}^{b}\right)$$

If the aggregation number of the ternary micelle tends to infinity, then the following limit value applies:(7)$$\underset{n\to \infty }{\lim}\ln{\left(x_{mM}^{b}\right)}^{\frac{1}{n}}=0$$

Therefore, Equation (6) is:(8)$$\frac{-\left(\mu_{mM}^{0}-n_{1}\mu_{1}^{0}-n_{2}\mu_{2}^{0}-n_{3}\mu_{3}^{0}\right)}{n}=-RT\left(x_{1}\ln x_{1}^{b}+x_{2}\ln x_{2}^{b}+x_{3}\ln x_{3}^{b}\right)$$
(9)$${{∆}_{f}g}_{mM}^{0}=RT\left(x_{1}\ln x_{1}^{b}+x_{2}\ln x_{2}^{b}+x_{3}\ln x_{3}^{b}\right)$$

In Equation (9), $x_{i}^{b}$ (i=1,2,3) can be expressed as products: critical micellar concentrations of the ternary mixture of the surfactant (${CMC}_{123}$) and the molar fraction of the corresponding surfactant from the binary mixture of surfactants ($\alpha_{i}$):(10)$${{∆}_{f}g}_{mM}^{0}=RT\left(x_{1}\ln\alpha_{1}{CMC}_{123}+x_{2}\ln\alpha_{2}{CMC}_{123}+x_{3}\ln\alpha_{3}{CMC}_{123}\right)$$

If the limiting value (7) is valid, formation of the ternary mixed micelle can equally be described using the law of mass action (i.e., association reaction (4)) and the phase separation method. In the state of equilibrium, the chemical potential of the surfactant from the ternary mixed micellar pseudophase is equal to the chemical potential of the same surfactant from the aqueous phase [14,15,16]:(11)$$\overset{mixed\;micellar\;pseudophase}{\overbrace{\underset{\mu_{mm}^{0}}{\underbrace{\mu_{\left(aq\right)}^{0}+RTln{CMC}_{i}}}+RTlnx_{i}f_{i}}}=\overset{water\;phase}{\overbrace{\mu_{\left(aq\right)}^{0}+RTln\alpha_{i}{CMC}_{123}}},\;(i=1,2,3)$$

From the equality of chemical potentials follows the expression (11):(12)$${CMC}_{i}x_{i}f_{i}=\alpha_{i}{CMC}_{123},\;(i=1,2,3)\;$$
where $f_{i}$ represents the coefficient of the activity of surfactants in ternary mixed micelle pseudophases (if the activity coefficient of a particle in a real mixture has a lower value than 1, then the observed particle is thermodynamically more stable than in a hypothetical ideal mixture under the same conditions [27]). By introducing expression (12) into Equation (10):(13)$$\frac{{{∆}_{f}g}_{mM}^{0}}{RT}=\overset{x_{1}{∆}_{f}g_{M1}^{0}+x_{2}{{∆}_{f}g}_{M2}^{0}+x_{3}{{∆}_{f}g}_{M3}^{0}}{\overbrace{x_{1}\ln{CMC}_{1}+x_{2}\ln{CMC}_{2}+x_{3}\ln{CMC}_{3}}}+\underset{{∆g}_{mix}^{id}}{\underbrace{x_{1}\ln x_{1}+x_{2}\ln x_{2}+x_{3}\ln x_{3}}}+\underset{g_{mix}^{E}}{\underbrace{x_{1}\ln f_{1}+x_{2}\ln f_{2}+x_{3}\ln f_{3}}}$$
we get Equation (13) that is equivalent to expression (1); of course, if the limit value (7) holds. Rearranging Equation (10) gives:(14)$${{∆}_{f}g}_{mM}^{0}=RT\left(x_{1}\ln{CMC}_{123}+x_{2}\ln{CMC}_{123}+x_{3}\ln{CMC}_{123}+x_{1}\ln\alpha_{1}+x_{2}\ln\alpha_{2}+x_{3}\ln\alpha_{3}\right)$$
and if the limiting value exists:(15)$$\underset{n\to \infty }{\lim}\ln\alpha_{i}^{\frac{1}{n}}=0,\;(i=1,2,3)$$
then Equation (14) is:(16)$${{∆}_{f}g}_{mM}^{0}=RT\left(\underset{1}{\underbrace{\left(x_{1}+x_{2}+x_{3}\right)}}\ln{CMC}_{123}+\underset{0}{\underbrace{n_{1}\ln\alpha_{1}^{\frac{1}{n}}+n_{2}\ln\alpha_{2}^{\frac{1}{n}}+n_{3}\ln\alpha_{3}^{\frac{1}{n}}}}\right)$$
(17)$${{∆}_{f}g}_{mM}^{0}=RT\ln{CMC}_{123}$$

Thus, if the aggregation number of the ternary mixed micelle is relatively high, then the same expression (17) is obtained for ${{∆}_{f}g}_{mM}^{0}$, starting from the association reaction (4) or from expression (1) in the phase separation theory (Figure 2). If classic surfactants with hydrocarbon chains form mixed micelles with bile acid anions, then molecular dynamic simulations show that the applicability of limit values (7) and (15) is justified [41]. In Equation (17), ${CMC}_{123}$ is expressed in mol·dm^−3^ or in the surfactant mole fraction. For surfactants with low ${CMC}_{123}$ values (<10 mM), the values of ${{∆}_{f}g}_{mM}^{0}$ would only differ by the constant term ≈ ln 55.5 = 4.02 when using one or the other unit [42]. 

In deriving Equation (17) from the associated reaction (4), the micelle charge and the binding of counterions to the micelle were not taken into account, i.e., partial neutralization of the micellar charge. The function of the dependence of the specific conductivity of the aqueous solution of the ternary mixture of the investigated surfactants on the concentration of the ternary mixture (Figure 3) (${CMC}_{123})$ was determined spectrofluorimetrically, with pyrene as the probe molecule (Table 1). This means that the examined ternary mixed micelle does not bind counterions to its outer shell, which is a consequence of the presence of bile acid anions (**3**), which, with their steroid skeleton on the surface of the micelle, disturb the continuous electrostatic potential of spheroidal symmetry and which exists on the surface of cationic monocomponent micelles. This then makes it difficult to form a Helmholtz layer in the Stern double layer—micellar building elements behave like individual surfactant particles in terms of charge, i.e., they are in a completely dissociated state [21,41]. 

In the examined ternary mixed micellar pseudophase, each surfactant on each temperature has the negative value of the logarithm of the coefficient of activity; also, the excess molar Gibbs energy is negative, which means that the real ternary mixed micellar pseudophase is thermodynamically more stable than the ideal micellar pseudophase (Table 1). By applying Nagarajan’s molecular theory of the micellar state, the following expression can be written for the excess molar Gibbs free energy [19]: (18)$$g_{mix}^{E}=x_{1}{\left(∆h_{ord}-RT\ln\frac{\Omega_{mM}}{\Omega_{M}}\right)}_{1}+x_{2}{\left(∆h_{ord}-RT\ln\frac{\Omega_{mM}}{\Omega_{M}}\right)}_{2}+\overset{0}{\overbrace{x_{3}{\left(∆h_{ord}-RT\ln\frac{\Omega_{mM}}{\Omega_{M}}\right)}_{3}}}\;+\gamma \left(a-x_{1}a_{1}-x_{2}a_{2}-x_{3}a_{3}\right)+x_{1}∆g_{dipol}+x_{2}∆g_{dipol}+∆g_{Columb}$$

The first three terms present the change in enthalpy ($∆h_{ord}$) and entropy (expressed over the number of microstates in the mixed micelle ($\Omega_{mM}$) and in monocomponent micelle ($\Omega_{M}$)) for each surfactant due to the change in the conformational states of the hydrocarbon chain in the mixed micelle compared to the conformations in monocomponent micelles [38]. As the anion of deoxycholic acid (**3**) contains a conformationally rigid steroid skeleton in its structure, the conformation of the anion of deoxycholic acid is independent of the micellar packing, i.e., the structure of the neighboring surfactant and the coordination number of the micellar pseudophase. The fourth term of the expression (18) originates from the fact that the effective surface area of the hydrophobic core of the ternary mixed micelle (a) is not a linear combination of the product of the mole fractions of the surfactants and the effective surface area of the hydrophobic cores of monocomponent micelles ($x_{i}a_{i},\;(i=1,2,3)$) [19]. The fifth term ($x_{1}∆g_{dipol}+x_{2}∆g_{dipol}$) relates to the cation– dipole interaction between cationic surfactants and the C3 pseudoaxial and C12 axial OH groups of the deoxycholic acid anion (**3**) [22]. The last term takes into account the reduction in repulsive electrostatic interactions between identical charges in monocomponent micelles during the formation of ternary mixed micelles with attractive electrostatic interactions between anionic and cationic surfactants.

In the system of three structurally different surfactants in the aqueous solution, surfactant (**1**) is the most hydrophobic (has the lowest critical micellar concentration, Figure 1 and Appendix B); thus, surfactant (**1**) probably first forms a monocomponent micelle in which there are particles (**1**) with elongated hydrocarbon chains and more or less globular conformations [43] (in order to evenly fill the micellar core). The deoxycholic acid anion incorporates in the micelle by substituting surfactant (**1**) in globular conformation—the steroid skeleton of surfactant (**3**) is localized in the micellar groove, i.e., between elongated conformations of surfactant (**1**) as the convex surface (β side of the steroid skeleton with angular methyl groups) particles (**3**) are oriented towards the interior of the micelle, while the axial OH groups and the C17 side chain with carboxylate group towards the aqueous solution orientation of the deoxycholic acid anion is confirmed by cross peaks in a 2D ROESY [44,45] experiment (Figure 4 and Figure 5). Surfactant (**2**), the least hydrophobic surfactant in the examined mixture, probably incorporates with the elongated conformations in the micelle of surfactants (**1**) and (**3**).

Therefore, in ternary micelles of cationic surfactants, (**1**) and (**2**) decrease the number of conformational microstates related to the monocomponent micelles; so, in the expression (18), the first and the second term are:(19)$$x_{1}{\left(∆h_{ord}-RT\ln\frac{\Omega_{mM}}{\Omega_{M}}\right)}_{1}+x_{2}{\left(∆h_{ord}-RT\ln\frac{\Omega_{mM}}{\Omega_{M}}\right)}_{2}>0$$
i.e., the effect of reducing the number of conformational microstates in ternary mixed micelles thermodynamically destabilizes the real mixed micelle in relation to the state of the ideal mixed micelles. In monocomponent micelles of surfactants (**1**) and (**2**), when the hydrocarbon chain of cationic surfactant is in globular conformation, the hydrophobic molecular surface is exposed to hydration [43]. The substitution of globular conformations of cationic surfactants with deoxycholic acid anions results in a decrease in the degree of hydrophobic hydration since the effective hydrophobic surface of the ternary mixed micelle decreases compared to monocomponent micelles (Figure 5). Thus, the fourth term in the expression (18) is:(20)$$\gamma \left(a-x_{1}a_{1}-x_{2}a_{2}-x_{3}a_{3}\right)<0$$
and thermodynamically stabilizes real mixed micelles towards the ideal mixed micelles. Since the intermolecular interactions between the dipole OH surfactant (**3**) and cationic surfactants (**1**) and (**2**) do not exist in monocomponent micelles (**1**) and (**2**), the last terms of expression (18) have a stabilizing contribution to the ternary mixed micelle compared to the ideal ternary mixed micelle (Figure 6):(21)$$x_{1}∆g_{dipol}+x_{2}∆g_{dipol}+∆g_{Columb}<0$$

The ternary mixed micelle’s excess molar Gibbs free energy is $g_{mix}^{E}<0$ (Table 1), hence, it must be: (22)$$\left|x_{1}{\left(∆g_{ord}\right)}_{1}+x_{2}{\left(∆g_{ord}\right)}_{2}\right|<\left|\gamma \left(a-x_{1}a_{1}-x_{2}a_{2}-x_{3}a_{3}\right)+x_{1}∆g_{dipol}+x_{2}∆g_{dipol}+∆g_{Columb}\right|$$
where $∆g_{ord}=∆h_{ord}-RTln(\Omega_{mM}/\Omega_{M})$, ord = ordering.

In the case of a ternary mixture of surfactants with the following composition: α_1_ = 0.05; α_2_ = 0.35; α_3_ = 0.6 (Table 1), in the formed ternary mixed micellar pseudophase, the molar fractions of cationic surfactants are approximately equal to each other; although, in the initial mixture, the amount of cationic surfactants with a shorter hydrocarbon chain (**2**) is seven times greater than the amount of cationic surfactants with a longer hydrocarbon chain (**1**).

In binary micellar pseudophases, both cationic surfactants have synergistic interactions of similar strengths with the deoxycholic acid anion (Appendix B). This means that in the formation of the ternary mixed micellar pseudophase system (i.e., surfactant aqueous solution), the more hydrophobic surfactant (**1**) that is incorporated into the mixed micelle, the micelle is more stabilized. In this way, a larger amount of water molecules from the hydration layer above the hydrophobic molecular surface of the monomeric surfactant (water molecules that have a lower entropy compared to water molecules from the interior of the aqueous solution [46]) move into the interior of the aqueous solution; thus, the entropy of the system increases (especially at lower temperatures [47,48,49,50,51,52]). The relation $g_{mix}^{E}/{∆}_{f}g_{mM}^{0}$ grows with the temperature rise (Table 1), since with increasing temperatures the difference in entropy between water molecules in the hydration layer above the hydrophobic molecular surface and water molecules inside the aqueous solution decreases [46], i.e., in Equation (1), the absolute value of first three terms decreases ($x_{1}{∆}_{f}g_{M1}^{0}+x_{2}{{∆}_{f}g}_{M2}^{0}+x_{3}{{∆}_{f}g}_{M3}^{0}$), in which, according to Nagarajan and Tanford, the entropy change due to the transfer of the hydrophobic molecular segment from the aqueous phase to the hydrophobic environment is incorporated (along with the dehydration of the hydrophobic surface) [19]. However, since $g_{mix}^{E}$ contains electrostatic attractive interactions between cationic and anionic surfactants that do not change with the temperature, the relation $g_{mix}^{E}/{∆}_{f}g_{mM}^{0}$ slightly decreases ($g_{mix}^{E}$ according to Equation (1) is incorporated in ${∆}_{f}g_{mM}^{0}$; both Gibbs energies slighly increase with temperature (Table 1) due to the increase in hydrophobic interactions (i.e., van der Waals interactions [46]) that do not exist in monocomponent micelles as they are not linear functions of hydrophobic interactions from monocomponent micelles). Similarly, $g_{mix}^{E}/{∆}_{f}g_{mM}^{0}$ changes in the case of the ternary mixtures of the aqueous solutions of surfactants of the composition α_1_ = 0.1; α_2_ = 0.3; α_3_ = 0.6 (Table 1). However, in this case with a temperature rise, it decreases the molar fraction of surfactant (**1**), while the molar fraction of surfactant (**2**) grows in the micellar pseudophase. At higher temperatures, surfactant (**1**) probably takes some globular conformations as well, so the surface below globular surfactant (**1**) easily (without steric repulsive interactions) fills with surfactants with shorter hydrocarbon chains. For other examined mixtures, the relation $g_{mix}^{E}/{∆}_{f}g_{mM}^{0}$ does not change with the temperature, but remains more or less constant (with some fluctuation). Namely, with the increase in α_1_ in the ternary mixed micellar pseudophase, the probability increases, especially at higher temperatures, so that a certain fraction of surfactant (**1**) receives a more or less globular conformation, whereby in the hydrocarbon chains of surfactant (**1**), partial synclinal and synperiplanar conformations are formed (i.e., conformations with repulsive interactions). [38]. Thus, the part of the electrostatic attractive interaction (energy) is used for achieving certain conformation states (especially at higher temperatures) [21], which results in a more or less constant value of the relation $g_{mix}^{E}/{∆}_{f}g_{mM}^{0}$ in the temperature dependence. 

## 3. Materials and Methods

All chemicals were used from the original manufacturer’s packaging (Table 2).

### 3.1. Determinations of Critical Micellar Concentration

Stock solutions of surfactants (hexadecyltrimethylammonium bromide (**1**), dodecyltrimethylammonium bromide (**2**), sodium deoxycholate (**3**)) prepared in deionized water were mixed in the ternary mixtures with different molar ratios of surfactants (0.5:3.5:0.6, 1:3:6, 2:2:6, 3:1:6, 0.35:0.05:0.6). In the ternary mixture of the ratio 0.5:3.5:0.6, the ratio between surfactants 1:2 is 0.125:0.875; 1:3 is 0.077:0.923, and 2:3 is 0.368:0.632; these binary mixtures of surfactants were also measured in order to calculate the interaction factors of surfactants in the binary mixtures that were used for calculations in the ternary mixture. Accordingly, in Table 3, all the measured ternary and binary mixtures of surfactants are presented.

Critical micellar concentrations of monocomponent surfactants and their secondary and ternary mixtures were measured spectrofluorimetrically on Cary Eclipse fluorescence spectrophotometer (Agilent, Waldbronn, Germany) using pyrene as the probe molecule. The ratio of the pyrene fluorescence intensities of the first and the third vibronic peaks (a measure of the environmental polarity) was measured in the function of total concentration of surfactant on temperatures (278.1–313.1 K, in intervals of 5 K; the temperature variation is 0.1 K) and critical micelle concentrations were determined by curve fitting with the Boltzmann equation (OriginaLab 9 software) (Appendix C). Physico-chemical parameters of examined surfactant mixtures were calculated using Mathlab.

Thermodynamic parameters of ternary mixed micelle (Table 1) are calculated according to RST protocol [11,32,33] from data (coefficients of interaction) of binary systems (Appendix B). 

### 3.2. Conductivity Measurements

The goal of the conductometry measurement was to determine the fraction of counter ion binding to the mixed micelle. Conductivity was measured by gradual dilution of surfactant mixture solutions with deionized water. The data were acquired using a Consort C 860 conductometer. Equipment was calibrated with KCl solution ranging from 0.01 to 1.0 mol·dm^−3^ of known *κ* (specific conductivity). The cell containing solutions was immersed in a water bath, controlling the temperature variation at 0.1 K. The temperature was kept constant at 293.1 K.

### 3.3. Conductivity Measurements

For the NMR experiments, samples were prepared as D_2_O solution and 0.7 mL of solution were used for the measurement. Spectra were recorded on a Bruker AVANCE III HD 400 MHz spectrometer, equipped with Prodigy cooled probe head. For ROESY experiments, standard Bruker pulse program with water suppression (roesyphpr.2) was used with the spin-lock pulse length of 350 ms. 

## 4. Conclusions

The examined ternary mixed micelle is thermodynamically more stable than a hypothetical ideal mixed micelle ($g_{mix}^{E}<0$). Thermodynamic stabilization is most likely a result of electrostatic attractive interactions between cationic surfactants and anions of deoxycholic acid (**3**), as well as ion–dipole interactions involving the OH groups of the surfactant (**3**). If a ternary mixture has an α_1_ lower than 0.2, then the $g_{mix}^{E}/{∆}_{f}g_{mM}^{0}$ grows with the temperature; if α_1_ is higher than 0.2, then the ratio of Gibbs free energies $g_{mix}^{E}/{∆}_{f}g_{mM}^{0}$ is more or less constant. 

## Figures and Tables

**Figure 1 molecules-28-06722-f001:**
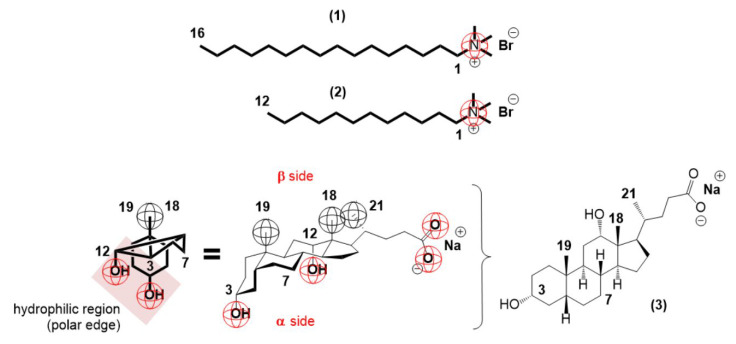
Surfactants: hexadecyltrimethylammonium bromide (**1**), dodecyltrimethylammonium bromide (**2**), and sodium deoxycholate (**3**).

**Figure 2 molecules-28-06722-f002:**
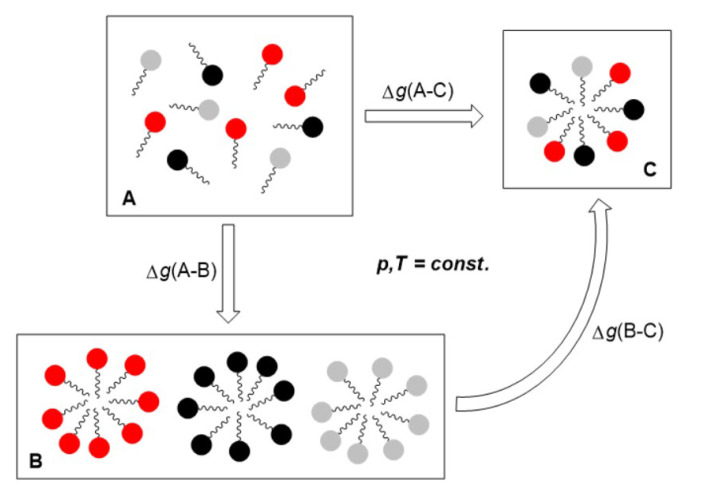
Hess’s law for formation of ternary mixed micelles: structurally different surfactants as monomers from aqueous solution (**A**) form monocomponent micelles as separated monocomponent micellar pseudophases (**B**) from which ternary mixed micellar pseudophase (**C**) is formed by mixing; process between states (A) and (B) corresponds to the direct association of surfactant monomers ($x_{1}{∆}_{f}g_{M1}^{0}+x_{2}{{∆}_{f}g}_{M2}^{0}+x_{3}{{∆}_{f}g}_{M3}^{0}=∆g(A-B$); ${∆g}_{mix}^{id}+g_{mix}^{E}=∆g(B-C)$ and ${{∆}_{f}g}_{mM}^{0}=∆g(A-C)$).

**Figure 3 molecules-28-06722-f003:**
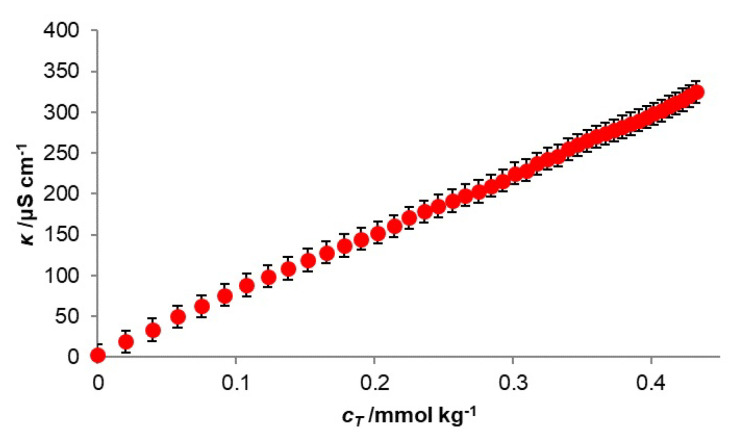
Change of specific conductivity as function of total surfactant mixture concentration (*T* = 293.1 K, *α*_1_ = 0.1, *α*_2_ = 0.3, *α*_3_ = 0.6).

**Figure 4 molecules-28-06722-f004:**
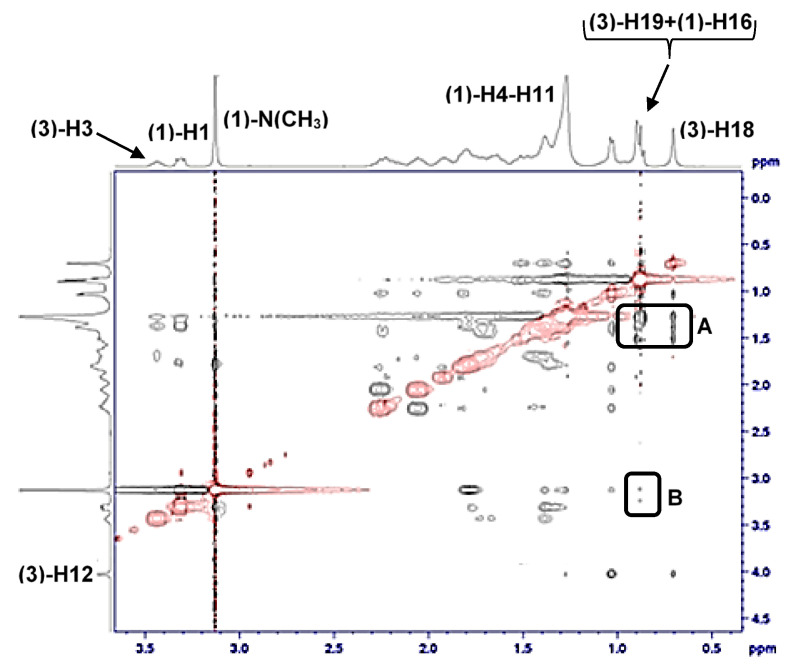
2D ROESY spectrum of binary mixed micelles (**1**)–(**3**) in 1: 1 molar ratio in aqueous solution above critical micellar concentration; *T* = 293.1 K: there are cross-peaks between the proton group from surfactant (**1**) (protons from the C4-C11 hydrocarbon segment methylene groups) and the proton group from (**3**) surfactant from the C21 steroid skeleton side chain methyl group and protons from C18 and C19 angular methyl groups of the steroid ring system—region A; in region B, there are cross peaks between C21 protons from (**3**) and protons from methyl groups bound to the (**1**) surfactant quaternary nitrogen atom.

**Figure 5 molecules-28-06722-f005:**
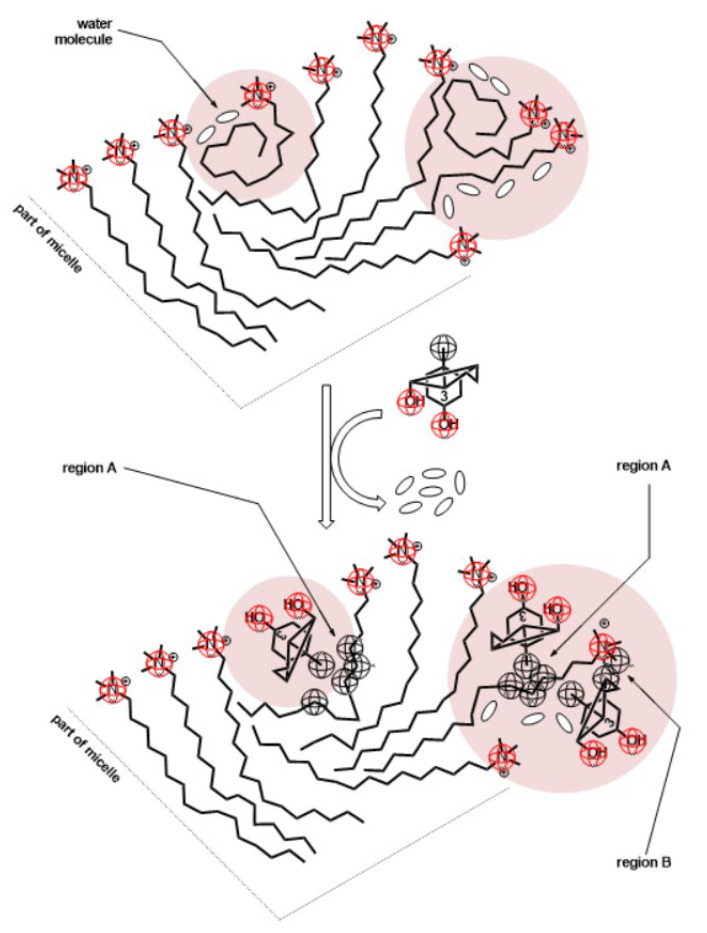
Cross-section of the mixed micelle: deoxycholic acid anion (**3**) replaces globular conformations of cationic surfactant; region A and B present segments of hydrocarbon chain whose protons give cross peaks in 2D ROESY spectra.

**Figure 6 molecules-28-06722-f006:**
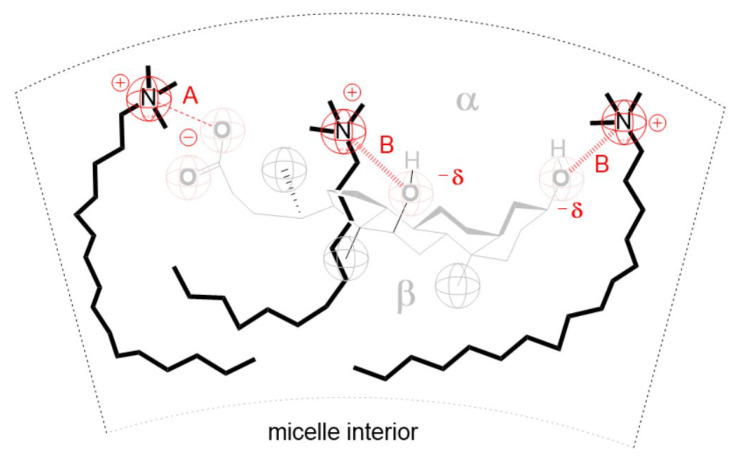
Intermolecular interactions in ternary mixed micelles that do not exist in monocomponent micelles: A- cation–anion attractive interaction and B- dipole–cation attractive interaction.

**Table 1 molecules-28-06722-t001:** Critical micelle concentrations, composition, coefficients of activity, and thermo-dynamic parameters of ternary mixed micelle hexadecyltrimethylammonium bromide (**1**)—dodecyltrimethylammonium bromide (**2**)—sodium deoxycholate (**3**).

*T* /K	CMC_123_ /mmolkg^−1^	*x* _1_	*x* _2_	*x* _3_	ln*f*_1_	ln*f*_2_	ln*f*_3_	$g_{mix}^{E}$ /kJmol^−1^	${{∆}_{f}g}_{mM}^{0}$ (17)/kJmol^−1^	$\frac{g_{mix}^{E}}{{∆}_{f}g_{mM}^{0}}$
***α*_1_ = 0.05; *α*_2_ = 0.35; *α*_3_ = 0.6**										
278.1	0.5128	0.33	0.28	0.39	−2.64	−2.56	−2.85	−6.25	−17.52	0.35
283.1	0.5362	0.32	0.29	0.39	−2.53	−2.53	−2.81	−6.22	−17.73	0.35
288.1	0.5471	0.31	0.29	0.39	−2.55	−2.56	−2.76	−6.31	−17.99	0.35
293.1	0.5362	0.31	0.29	0.40	−2.64	−2.66	−2.73	−6.54	−18.35	0.36
298.1	0.5595	0.30	0.31	0.39	−2.58	−2.74	−2.81	−6.74	−18.56	0.36
303.1	0.5704	0.30	0.31	0.39	−2.78	−2.72	−2.81	−6.99	−18.82	0.37
308.1	0.6156	0.31	0.29	0.39	−2.78	−2.85	−2.82	−7.22	−18.94	0.38
313.1	0.7307	0.32	0.31	0.38	−2.62	−2.86	−3.03	−7.41	−18.80	0.39
***α*_1_ = 0.1; *α*_2_ = 0.3; *α*_3_ = 0.6**										
278.1	0.2251	0.33	0.15	0.52	−2.93	−3.08	−1.72	−5.36	−19.42	0.27
283.1	0.2392	0.30	0.17	0.53	−2.73	−3.06	−1.51	−5.02	−19.63	0.26
288.1	0.2607	0.29	0.18	0.53	−2.63	−3.06	−1.49	−5.03	−19.76	0.25
293.1	0.2885	0.20	0.25	0.56	−1.90	−3.30	−1.31	−4.69	−19.86	0.23
298.1	0.3856	0.21	0.27	0.52	−1.80	−3.10	−1.56	−5.02	−19.48	0.26
303.1	0.4296	0.22	0.27	0.51	−1.95	−2.99	−1.59	−5.16	−19.54	0.26
308.1	0.5107	0.22	0.29	0.49	−1.72	−3.10	−1.93	−5.68	−19.42	0.29
313.1	0.5881	0.23	0.29	0.48	−1.68	−3.06	−2.00	−5.82	−19.36	0.30
***α*_1_ = 0.2; *α*_2_ = 0.2; *α*_3_ = 0.6**										
278.1	0.1811	0.58	0.11	0.31	−1.71	−1.57	−4.58	−5.96	−19.92	0.30
283.1	0.1425	0.57	0.14	0.29	−1.59	−1.45	−4.75	−5.84	−20.84	0.25
288.1	0.1858	0.57	0.11	0.33	−1.83	−1.66	−4.32	−6.28	−20.58	0.31
293.1	0.2129	0.51	0.02	0.46	−2.82	−2.65	−2.86	−6.91	−20.60	0.34
298.1	0.2455	0.49	0.01	0.50	−3.24	−3.06	−2.51	−7.14	−20.60	0.35
303.1	0.2843	0.46	0.05	0.49	−3.26	−2.99	−2.49	−7.22	−20.57	0.35
308.1	0.3225	0.39	0.12	0.48	−2.66	−3.13	−2.24	−6.44	−20.59	0.31
313.1	0.3772	0.35	0.14	0.51	−2.23	−3.15	−1.83	−5.61	−20.52	0.27
***α*_1_ = 0.3; *α*_2_ = 0.1; *α*_3_ = 0.6**										
278.1	0.2208	0.35	0.16	0.48	−3.33	−2.35	−2.92	−6.88	−19.46	0.35
283.1	0.2312	0.35	0.16	0.49	−3.53	−2.32	−2.95	−7.18	−19.71	0.36
288.1	0.2444	0.32	0.19	0.48	−3.42	−2.30	−2.88	−7.04	−19.92	0.35
293.1	0.2607	n.d.	n.d.	n.d.	n.d.	n.d.	n.d.	n.d.	−20.11	n.d.
298.1	0.2739	0.39	0.11	0.50	−3.34	−2.49	−2.87	−7.47	−20.33	0.37
303.1	0.2587	0.39	0.12	0.49	−3.15	−2.60	−2.68	−7.19	−20.81	0.35
308.1	0.4113	n.d.	n.d.	n.d.	n.d.	n.d.	n.d.	n.d.	−19.97	n.d.
313.1	0.4757	0.31	0.23	0.46	−2.31	−2.22	−2.97	−6.75	−19.92	0.34
***α*_1_ = 0.35; *α*_2_ = 0.05; *α*_3_ = 0.6**										
278.1	0.2272	0.55	0.09	0.35	−3.45	−3.10	−4.76	−8.98	−19.39	0.46
283.1	0.1991	0.55	0.03	0.42	−2.95	−2.55	−3.93	−7.88	−20.06	0.39
288.1	0.2149	0.60	0.02	0.38	−3.14	−2.80	−4.63	−8.86	−20.23	0.44
293.1	0.2557	0.61	0.02	0.37	−2.99	−2.72	−4.67	−8.79	−20.15	0.44
298.1	0.2594	0.59	0.03	0.38	−2.80	−2.63	−4.24	−8.30	−20.46	0.41
303.1	0.2865	0.50	0.11	0.39	−3.36	−3.07	−3.73	−8.75	−20.56	0.43
308.1	0.2737	0.54	0.03	0.43	−2.99	−3.01	−3.28	−7.98	−21.01	0.38
313.1	0.3805	0.54	0.01	0.45	−3.03	−2.57	−3.08	−7.93	−20.49	0.39

Molar Gibbs energy of formation of ternary micellar pseudophase (${∆}_{f}g_{mM}^{0}$) is calculated from Equation (17); α*_i_* presents molar fraction of surfactants in starting mixtures of surfactants dissolved in aqueous solution; the relative standard uncertainty of the critical micelle concentrations 4%.

**Table 2 molecules-28-06722-t002:** The origin and information of used chemicals.

Compound	Origin	CAS Number	Purity
Hexadecyltrimethylammonium bromide	Alfa Aesar	57-09-0	>0.980
Dodecyltrimethylammonium bromide	Alfa Aesar	1119-94-4	>0.980
Sodium Deoxycholate	Alfa Aesar	302-95-4	>0.980
Pyrene	Aldrich	129-00-0	>0.980

**Table 3 molecules-28-06722-t003:** Molar ratios of measured mixtures of surfactants hexadecyltrimethylammonium bromide (**1**)—dodecyltrimethylammonium bromide (**2**)—sodium deoxycholate (**3**); ternary mixtures and corresponding secondary mixtures.

Ternary Mixture 1:2:3	Corresponding Binary Mixtures		
	1:2	1:3	2:3
0.5:3.5:0.6	0.125:0.875	0.077:0.923	0.368:0.632
1:3:6	0.25:0.75	0.143:0.857	0.33:0.67
2:2:6	0.5:0.5	0.25:0.75	0.25:0.75
3:1:6	0.75:0.25	0.33:0.67	0.25:0.75
3.5:0.5:6	0.875:0.125	0.368:0.632	0.125:0.875

## Data Availability

Not applicable.

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
