# Peer review of "Ternary Mixed Micelle Hexadecyltrimethylammonium Bromide—Dodecyltrimethylammonium Bromide—Sodium Deoxycholate: Gibbs Free Energy of Mixing and Excess Gibbs Energy of Mixing"

_molecules, 2023, doi:10.3390/molecules28186722_

Round 1
Reviewer 1 Report
Posa et al. studied the micelle formation in the mixture of three different surfactants in aqueous solution, with theoretical background. The Introduction is nice, and the data was presented in detail in a few tables. While this is a piece of serious work, the referee would like to bring up the following issues:
(1) All studies are based on aqueous solutions of surfactants. It is well known that at higher temperatures, elevated hydrophobicity will make surfactants less soluble, leading to lower CMC and higher association numbers. However, in Table 1 and Table B1, there are multiple examples that for the same combinations of surfactants, the CMC increases with increasing temperature. This cannot be valid. Authors need to clarify.
(2) For the studies (such as those outlined in Table 1), the fraction of the third component was fixed at 0.6. This limits the impacts of the study as in such a case the third surfactant is acting as a background. Any possibility of explore the effect of this surfactant?
(3) While CMC values are reported, two other important parameters of micelles, total aggregation number and the fractions of three different surfactants in the mixed micelles, are missing. Any chance to get that information and show how do they change with the compositions of the mixed surfactants in solution, and temperature?
Author Response
The authors are very grateful for the referee comments.
1. Critical micelle concentrations of ionic surfactants, such as bile acid anions (bile salts), sodium dodecylsulfate, cetyltrimethylammonium bromide, etc., show temperature dependence whose function is U-shaped. On a certain temperature, TH micelle formation is exclusively entropic in nature- entropy driven T = (293-298) K, i.e. the change in the enthalpy of micellisation is zero, so on this temperature is the highest hydrophobic effect (the passage of water molecules from the hydration layer above the hydrophobic surface of the monomeric surfactant into the interior of the aqueous solution). As the temperature increases, the entropy difference between water molecules from the bulk of the aqueous solution and water molecules from the hydration layer above the hydrophobic molecular surface decreases, which means that with an increase in temperature, the change (increase) in entropy of micellization decreases (the entropic effect of micellization decreases), i.e. the tendency towards self-association decreases, which manifests itself with an increase in the critical micellar concentration with temperature (this is in agreement with the results from Table 1 and Tables from Appendix). There is also a temperature (T > 350 K) TS, on which the change in the entropy of micellisation is zero, the smallest hydrophobic effect, and micellisation is the result of the entalphic effect – enthalphy becomes the major driving force for aggregation (hydrophobic interaction). Hydrophobic interaction is the result of the induced dipol interactions between hydrophobic surfaces of the micelle building units in the micelle core. Deviations are possible if the aquoeous solution of surfactant contains certain additives that destrub the structure of the bulk water.
- Garidel, P.; Hildebrand, A. Thermodynamic properties of association of colloids. Therm. Anal. Calorim. 2005, 82, 483–489.
- Garidel, P.; Hildebrand, A.; Neubert, R.; Blume, A. Thermodynamic characterization of bile salt aggregation as a function of temperature and ionic strength using isotermal titration calorimetry. Langmuir 2000, 16, 5267–5275. https://doi.org/10.1021/la9912390.
- Paula, S.; Süs, W.; Tuchtenhagen, J.; Blume, A. Thermodynamics of micelle formation as a function of temperature: A high sensitivity titration calorimetry study. Phys. Chem. 1995, 99, 11742–11751. https://doi.org/10.1021/j100030a019.
- Vázquez-Gómez, S.; Pilar Vázquez-Tato, M.; Seijas, J.A.; Meijide, F.; de Frutos, S.; Vázquez Tato, J. Thermodynamics of the aggregation of the bile anions of obeticholic and chenodeoxycholic acids in aqueous solution. Mol. Liq. 2019, 296, 112092. https://doi.org/10.1016/j.molliq.2019.112092.
- Kroflic, A.; Sarac, B.; Bester-Rogac, M. Thermodynamic characterization of 3-[(3- cholamidopropyl)-dimethylammonium]-1-propanesulfonate (CHAPS) micellization using isothermal titration calorimetry: Temperature, salt, and pH dependence. Langmuir 2012, 28, 10363–10371. https://doi.org/10.1021/ la302133q.
- Mahbub, S.; Rub, M.A.; Hoque, Md.A.; Khana, M.A.; Asiri, A.M. Critical Micelle Concentrations of Sodium Dodecyl Sulfate and Cetyltrimethylammonium Bromide Mixtures in Binary Mixtures of Various Salts at Different Temperatures and Compositions. Russian Journal of Physical Chemistry A 2019, 93, 2043–2052. https://doi.org/10.1134/S0036024419100170.
2. Due to the pharmaceutical formulation α=0.6 is constant (it is explained in the Introduction: 119 -128). From preliminary research, we know that with an increase in the α value of the anion of deoxycholic acid (above α=0.6), the mole fraction of dodecyltrimethylammonium bromide (2) in the mixed micelle decreases, and practically the ternary micelle becomes a binary micelle of hexadecyltrimethylammonium bromide (1) and sodium deoxycholate (3).
- The main task was the thermodynamic characterization of ternary micelles in terms of thermodynamic stabilization of real mixed micelles in relation to the ideal micellar state, which does not require aggregation numbers. Mole fractions of the micellar building units of the ternary micelle are given in Table 1.
Reviewer 2 Report
Mixed cat-anionic surfactant systems have found widespread use and understanding of the thermodynamics of mixing is important from fundamental and practical point of view, so data reported here will attract the attention of scientists, university professors and students who know the basics of physical chemistry and interested in pharmacy and cosmetics. Mixing the ionic surfactant with two oppositely charged surfactants without common ion in the solvent (water) then results in a six-component system in a thermodynamic analysis, the phase behavior of which can not be easily investigated even at constant temperature and pressure.
There is one comment to the authors of the paper: When two oppositely charged ionic surfactants (two cationic and one anionic, as considered here) are mixed in non-stoichiometric ratios (2:3, as reported), there is still a tendency to associative phase separation with the formation of stoichiometric neutral cat-anionic solids. The reader will want to see the evidence that the system in question is indeed micellar single phase.
Author Response
The authors are very grateful for the referee's comments.
1. Each tested aqueous solution of the ternary mixture of surfactants was transparent, without sedimentation or cloudiness. This is confirmed by the Boltzmann functions of the dependence of the ratio of the pyrene fluorescence intensities of the first and the third vibronic peaks from the concentration of the ternary mixture of surfactants. If there is turbidity in the aqueous solution, then the Boltzmann functions cannot be obtained.
Round 2
Reviewer 1 Report
Now the manuscript is acceptable.